# Hate and Toxic Speech Detection in the Context of Covid-19 Pandemic using XAI: Ongoing Applied Research

**David Hardage**
The University of Texas at San Antonio
david.hardage@my.utsa.edu

**Paul Rad, PhD**
The University of Texas at San Antonio
paul.rad@utsa.edu

## Abstract

As social distancing, self-quarantines, and travel restrictions have shifted a lot of pandemic conversations to social media so does the spread of hate speech. While recent machine learning solutions for automated hate and offensive speech identification are available on Twitter, there are issues with their interpretability. We propose a novel use of learned feature importance which improves upon the performance of prior state-of-the-art text classification techniques, while producing more easily interpretable decisions. We also discuss both technical and practical challenges that remain for this task.

## 1 Introduction

In the day and age of social media, a person's thoughts and feelings can enter the public discourse at the click of a mouse or tap of a screen. With billions of individuals active on social media, the task of finding reviewing and classifying hate speech online quickly grows to a scale not achievable without the use of machine learning. Additionally, the definition of hate-speech can be broad and include many nuances, but in general hate speech is defined as communication which disparages or incites violence towards an individual or group based on that person or groups' cultural/ethnic background, gender or sexual orientation. (Schmidt and Wiegand, 2017). In the context of Covid-19, the United Nations has released guidelines on Covid-19 related hatespeech Guidance on COVID-19 related Hate Speech cautioning that Member States and Social Media companies that with the rise of Covid-19 cases there has also been an increase of hate speech. The UN warns that such communication could be used for scapegoating, stereotyping, racist and xenophobic purposes.

---

[1] UN Guidance Note on Addressing and Countering COVID-19 related Hate Speech 11 May, 20

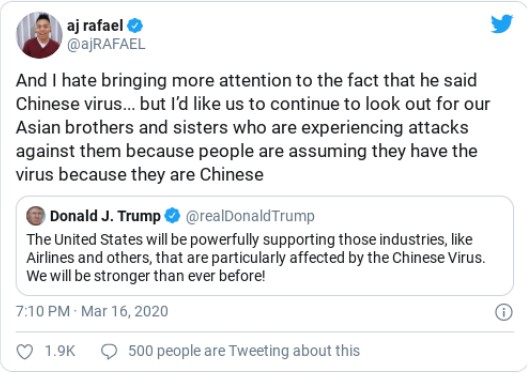

Figure 1: The tweets above displays an example of hate speech used for scapegoating by @realDonaldTrump and the response of @ajRAFAEL highlighting the impact this hate speech has on Asian Americans.

The importance of identifying hate speech combined with the magnitude of the data makes this an area in which innovations achieved in NLP and AI research can make an impact. However, the datasets we use reflect their environments and even their annotators (Waseem, 2016) (Sap et al., 2019), there are inherent cues contained by the data which can bias the predictions of models developed from these data (Davidson et al., 2019). In the context of detecting hate speech detection, this can lead to predictions be largely the outcome of a few key terms (Davidson et al., 2017). Being able to explain how underlying data impacts AI decision outcomes has real world applications, and social media companies ignorant to this fact could face a multitude of ethical and legal repercussions (Samek et al., 2017).

**Our Contribution:** In this research, we merge feature importance with text classification to help decrease false positives. Our method combines the global representation of a term's feature importance to a predicted class with the local term feature importance of an individual observation. Each term's

| | fears | of | visa | rejection | or | deportation | keep | immigrant | families | from | receiving | health | care | and | food | aid |
|---|---|---|---|---|---|---|---|---|---|---|---|---|---|---|---|---|
| Immigration Hate Speech Classification Local Feature Importance | fears | of | visa | rejection | or | deportation | keep | immigrant | families | from | receiving | health | care | and | food | aid |
| Immigration Hate Speech Classification Global Feature Importance | fears | of | visa | rejection | or | deportation | keep | immigrant | families | from | receiving | health | care | and | food | aid |
| No Hate Speech Classification Local Feature Importance | fears | of | visa | rejection | or | deportation | keep | immigrant | families | from | receiving | health | care | and | food | aid |
| No Hate Speech Classification Global Feature Importance | fears | of | visa | rejection | or | deportation | keep | immigrant | families | from | receiving | health | care | and | food | aid |
| Immigration Hate Speech Classification Global/Local Term Differences | 0.39 | 0.93 | 0.76 | -0.20 | 0.79 | 0.62 | 0.95 | 0.91 | 0.59 | 0.17 | -0.62 | 0.14 | 0.57 | 0.10 | 0.86 | 0.58 |
| No Hate Speech Classification Global/Local Term Differences | 0.91 | 0.87 | 0.83 | 0.38 | 0.92 | 0.89 | 0.92 | 0.87 | 0.88 | 0.74 | 0.74 | 0.66 | 0.98 | 0.99 | 0.98 | 0.32 |

Figure 2: This is an example of a Covid-19 Tweet incorrectly classified by our baseline model as "hate speech towards immigrants". After the applying our prediction enhancement method, the tweet was correctly classified as "not hate speech". The first two sentence combinations show differences in local and global term importance impacting the Term Difference Multiplier. The intensity of grey represents the importance of each term to the denoted label. The last sentence pair provides the term difference for each local and global term pair as described in our experimental design.

global feature importance is collected from our training dataset and baseline model. Then local feature importance is calculated for each observation on which our trained model makes a prediction. Our algorithm, uses the term level global feature importance to penalize model predictions when an observation's local term feature importance differs from the global feature importance.

## 2 Explainability and Text Classification

In the same vein of our research, others have leveraged explainability derived with integrated gradients (Sundararajan et al., 2017) and subject matter experts to create priors for use in text classification. In this research, they showed a decrease in undesired model bias and an increase in model performance when using scarce data (Liu and Avci, 2019). Overall, our method appears to have similar results and lessens the impacts of specific key terms to the the overall model prediction.

One of the more commonly utilized explainability methods, SHAP provides a framework within the feature contrubutions to a a model's output can be derived by borrowing Aultman-Shapely values from cooperative game theory (Lundberg and Lee, 2017). While there are several "explainer" implementations included with SHAP, Gradint Explainer allowed us to leverage our entire training dataset as the background dataset which allows our global average term values described in our experimental design to represent all terms in the training corpus.

SHAP's Gradient Explainer builds off of integrated gradients and leverages what are called expected gradients. This feature attribution method takes the integral from integrated gradients and reformulates it as an expectation usable in calculating the Shapely values. The resulting attributions sum to the difference between the expected and current model output. However, this method does assume independence of the input features, so it would violate this assumption if we were to leverage any sequence models in classification.

## 3 Datasets

### 3.1 Training and Evaluation Data

For this research, our intent to score unlabeled tweets called for a robust dataset which could be generalize to Covid-19 tweets. This lead us to combining three datasets in the domain of hate and offensive speech: the collection of racist and sexist tweets presented by Waseem and Hovy (Waseem and Hovy, 2016), the Offensive Language Identification Dataset (OLID) (Zampieri et al., 2019), and Multilingual Detection of Hate Speech Against Immigrants and Women in Twitter (HatEval)(Basile et al., 2019).

All hate or offensive labels contained within these three datasets were combined and given a sub-classification based on terms contained within each example. The sub-classes focus on hate and offensive speech targeting or directed towards immigrants, sexist, or political topics and/or individuals. The terms which divided positive labeled data into these three sub classes were derived through analysis of the terms contained in positive labeled tweets and through the terms used in extracting the tweets for the original data datasets.

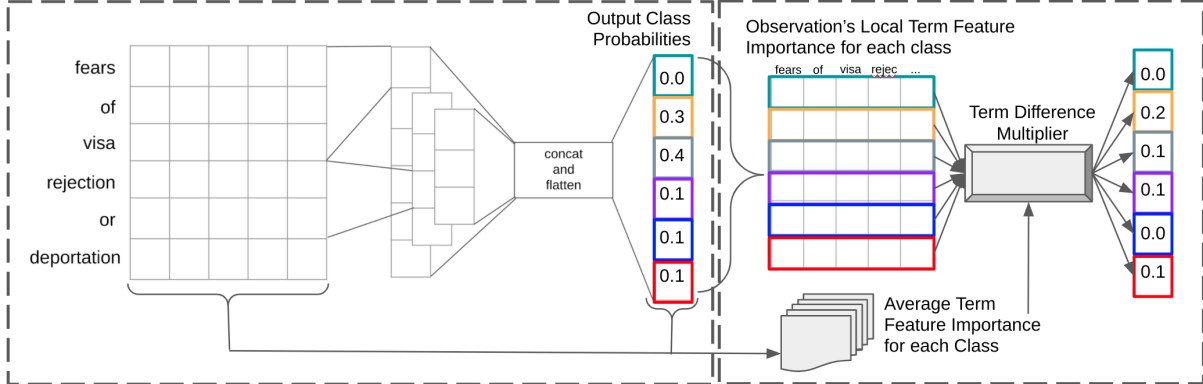

Figure 3: The architecture of both our baseline model and the prediction enhancement. Once the baseline model is trained we store the average term feature importance of terms in the training dataset to each predicted class. This is used as the global representation of that term's importance which we use with an individual observation's local term feature importance to derive the Term Difference Multiplier.

## 3.2 Covid-19 Tweets

In order to leverage this research in the context of Covid-19, we collected tweets using two different sources. First, we leveraged data collected by the Texas Advanced Computing Center (TAAC) at the University of Texas at Austin . This dataset was important for us to use due to "Chinese Virus" being one of the term pairs the TACC team used to collect data. In the context of hate speech in Covid-19, terms which target countries or ethnicity's in the labeling of the virus clearly disregard the aforementioned UN guidance on hate speech. The second dataset we leveraged was provided by Georgia State University's Panacea Lab (Banda et al., 2020) . For this dataset, we specifically hydrated tweets from the days following the murder of George Floyd and begging of civil unrest in America. The intent behind limiting to these dates was to increase the chance of capturing tweets containing racial or ethnic terms.

## 4 Experimental Design

### 4.1 Text Classifier

Since our research focus is to leverage learned feature importance to enhance predictions, we followed proven methods for hate speech classification (Gambäck and Sikdar, 2017). All tweets were converted into the Glove twitter embeddings (Pennington et al., 2014). These embeddings were passed to a Convolution Neural Network classifier

²https://www.tacc.utexas.edu/-/tacc-covid-19-twitter-dataset-enables-social-science-research-about-pandemic

which mirrored the same architecture used by Yoon Kim for sentence classification (Kim, 2014). We allowed for parameter tuning with random search, and the final CNN consisted of three convolutional layers of 75 filters with kernel sizes of 3, 4 and 6. These all received one dimensional max pooling and a dropout rate of 0.4 was applied. The output layer is a softmax with l2 regularization set at 0.029. These parameters were selected by ranking validation AUC. This model served as both our baseline model and the input predictions of our predictions enhanced with XAI.

### 4.2 Calculating Global Average Feature Importance

To achieve this, we apply SHAP's Gradient Explainer to our baseline model. The Gradient Explainer output has the same dimensions as our Glove embedded data, so to reduce the dimensionality to that of the input text sequence, we sum the expected gradients across the axis corresponding to a term in each sequence.

$$s_1 \left[ \sum_{x_1}^{x_n} x_i = x_1 + ... + x_n \right] \to \cdots \to s_n \left[ \sum_{x_1}^{x_n} \cdots \right]$$

Where s is a token in each sequence and $x_i$ is the summation of all expected gradients for the embedding dimensions. The values of these summations can be both positive and negative. Since our method requires positive inputs to measure the percentage difference, these values for every s step across all sequences in the training dataset are scaled between 0 and 1 via min/max scaling. We

then create a dictionary of terms from the training corpus and store the "global average" feature importance of each term. This dictionary of global average feature importance values is used to calculate how far a particular prediction strays from the feature importance represented in our training data.

### 4.3 Enhancing Predictions with Term Difference Multiplier

Now that we have the global importance of each term (feature token) to each class, we calculate the percentage difference of each term's local feature importance to that term's global feature importance by each class.

$$1 - \left( \frac{|s_g - s_l|}{\left\lceil \frac{(s_g + s_l)}{2} \right\rceil} \right)$$

As you can see above the percentage difference between each local ($s_l$) and global ($s_g$) feature importance is subtracted from 1. This outputs the difference multiplier for each term in a sequence. These values are averaged for each sequence, and the predicted probability for each class is multiplied by it's local Term Difference Multiplier for each sequence. This outputs a new predicted probability score which has been penalized based on how much it's local attribution values differ from the global mean of each term in the input sequence.

### 5 Results and Application to Covid 19 Tweets

We found that the enhanced predictions predominately help to correct false positive classifications and shift predictions towards the negative class. We hypothesize this is due to the diversity of language and relatively neutral feature importance most terms have on the negative class. The relative neutrality in both global and local feature importance scores can be seen in figure 2, and this results in a higher overall average for the aggregation of the Term Difference Multiplier.

When we applied our model to Covid-19 tweets we found similar results as described above. Quite often, we found that tweets providing information about specific ethnic groups or migrants were labeled as hateful or toxic towards immigrants by our baseline model and then correctly labeled as not hateful or offensive speech when we applied the Term Difference Multiplier. An example of this exact scenario is provided in figure 2.

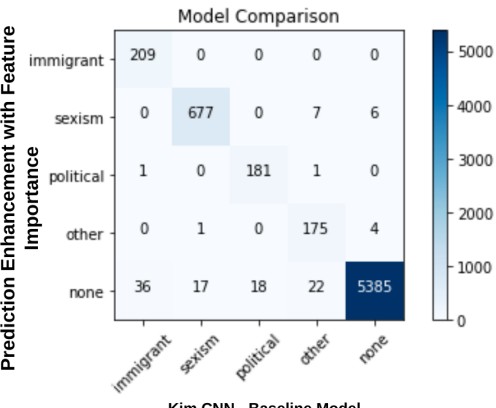

Figure 4: A comparison of the classification outputs from our two implementations. As you can see above, enhancing predictions with the Term Difference Multiplier shifts predictions to the majority class of no hate or toxic speech.

### 6 Conclusion and Future Work

Here we have experimented with a novel method to leverage the global feature importance from a model's training dataset to reinforce or even penalize new predictions when their local feature importance varies from this learned global value. This novel algorithm marries the field of XAI and NLP in a manner which allows prior knowledge obtained in model training to impact present predictions.

Overall, we believe this technique is especially applicable in scenarios like Covid-19 where little to no pre-existing labeled data are available. By training this method on a similar corpus it can be used to detract from incorrect predictions made due to a few highly influential terms in Covid-19 datasets. At present due to this method's ability to decrease false positives, we believe one application of this research is increasing the efficiency of systems monitoring for hateful and toxic communication. However, this research is ongoing. We intend to explore further scenarios such as altering the equation used in our Term Difference Multiplier and the datasets used since the global feature importance can greatly influence the multiplier combined with our model's original predicted probabilities.

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
