# OpenReview forum: "Hate and Toxic Speech Detection in the Context of Covid-19 Pandemic using XAI: Ongoing Applied Research"
_EMNLP/2020/Workshop/NLP-COVID — NLP-COVID19-EMNLP Poster_

### Official Review · AnonReviewer1 · 2020-09-10
**Interesting direction but could benefit from more rigor**

**Rating:** 4
**Confidence:** 3

**Review:**

This paper proposes a method for detecting hate speech that supposedly improves performance and explainability.
While the direction here is important and interesting, I think the paper has several key details that are missing and need to be included before it would be ready for publication.

First, if there is a quantitative evaluation of whether their proposed method improves performance, it was not clear to me. For example, having a Table with performances of the baseline versus their method (with respect to some ground truth) could clearly show the improvement in performance. Or plotting an ROC curve could be appropriate here
The datasets that were used are not clearly or adequately described. What is the ground truth and how were they annotated? What is the relationship between the training sets (which do not appear to be COVID-related) and the test set? Is there a data distribution mismatch due to the different topics?
Is there a quantitative way to measure explainability? All I see is one qualitative example.
Additionally, it seems that the authors proposed method shifts examples towards the "none" class, but it is unclear how this correlates with better performance. What's the ground truth?
I think what the authors mean to show in Figure 4 is that there are fewer false negatives. But was the number of false positives the same?
Authors should explain why they have chosen to use CNN instead of BERT as the text classifier.

---

### Official Review · AnonReviewer2 · 2020-09-22
**Very relevant application in need of more experiments and a rigorous evaluation**

**Rating:** 4
**Confidence:** 3

**Review:**

In this work the authors present their approach for hate and toxic speech detection in the context of the COVID-19 pandemic. For a short paper, the authors frame their solution well and discuss some of the relevant approaches by other researchers. One of the major drawbacks here is that almost no mention of transformer solutions, which have been more popular and better performing in the last few years. While the focus is put more on the contribution of using SHAP (SHapley Additive exPlanations) to focus on the importance of the features in the general context. The biggest issue with this paper comes with the lack of clear and concise experiments (baseline evaluations on the non-covid datasets), and a proper way of evaluating their methodology on the COVID-19 set, which of course has no gold-standard at the time.  While the opening of the paper is quite well crafted, the important evaluation and results section needs considerable work for the paper to be clear and fully self-standing.

Other items:
Sharing tweets like on the screenshot shown as figure 1 is definitely not ok in terms of privacy issues and it would go against Twitter's terms and conditions as well. A paraphrased summary of the tweet and removing the author would be the best way to share.

---

### Official Review · AnonReviewer3 · 2020-09-25
**Needs more work**

**Rating:** 4
**Confidence:** 4

**Review:**

This paper anchors on interpretability of machine learning solutions in the context of hate speech detection around COVID-19 social media. This problem is extremely crucial to address, especially in the current times. They leverage the global feature importance from a model’s training dataset to reinforce or penalize new predictions when their local feature importance varies from the learned global values.

The paper seems to be put in haste and lacks good presentation. Contributions section doesn’t clearly highlight why the proposed method is impactful. Results section feels incomplete due to lack of any rigorous evaluation. They should consider adding more qualitative samples to drive the point home on both the approach and interpretability aspects. The dataset collected in the context of COVID-19 (Section 3.2) deserve more analysis and isn’t described adequately. While leveraging “proven methods” is understandable, additional context would be helpful as to why other approaches were not considered or evaluated. Finally, I would suggest them to remove Figure 1. and consider providing multiple anonymized examples to illustrate the problem they are trying to solve.